# Conformational Rearrangements Regulating the DNA Repair Protein APE1

**DOI:** 10.3390/ijms23148015

**Published:** 2022-07-20

**Authors:** Nina Komaniecka, Marta Porras, Louis Cairn, Jon Ander Santas, Nerea Ferreiro, Juan Carlos Penedo, Sonia Bañuelos

**Affiliations:** 1Biofisika Institute (UPV/EHU, CSIC), University of the Basque Country (UPV/EHU), 48940 Leioa, Spain; n.komaniecka@o2.pl (N.K.); mporras001@ikasle.ehu.es (M.P.); louisjosephcairn@hotmail.com (L.C.); jonandersantas@gmail.com (J.A.S.); nferreiro002@ikasle.ehu.es (N.F.); 2Department of Biochemistry and Molecular Biology, University of the Basque Country (UPV/EHU), 48940 Leioa, Spain; 3Centre of Biophotonics, Laboratory for Biophysics and Biomolecular Dynamics, Scottish Universities Physics Alliance (SUPA) School of Physics and Astronomy, University of St. Andrews, St. Andrews KY16 9SS, UK; jcp10@st-andrews.ac.uk; 4Centre of Biophotonics, Laboratory for Biophysics and Biomolecular Dynamics, Biomedical Sciences Research Complex, School of Biology, University of St. Andrews, St. Andrews KY16 9ST, UK

**Keywords:** APE1, BER, DNA repair, fluorescence, FRET, NPM1, nucleophosmin, protein–DNA interaction, protein structure

## Abstract

Apurinic apyrimidinic endonuclease 1 (APE1) is a key enzyme of the Base Excision Repair (BER) pathway, which primarily manages oxidative lesions of DNA. Once the damaged base is removed, APE1 recognises the resulting abasic site and cleaves the phosphodiester backbone to allow for the correction by subsequent enzymes of the BER machinery. In spite of a wealth of information on APE1 structure and activity, its regulation mechanism still remains to be understood. Human APE1 consists of a globular catalytic domain preceded by a flexible N-terminal extension, which might be involved in the interaction with DNA. Moreover, the binding of the nuclear chaperone nucleophosmin (NPM1) to this region has been reported to impact APE1 catalysis. To evaluate intra- and inter-molecular conformational rearrangements upon DNA binding, incision, and interaction with NPM1, we used Förster resonance energy transfer (FRET), a fluorescence spectroscopy technique sensitive to molecular distances. Our results suggest that the N-terminus approaches the DNA at the downstream side of the abasic site and enables the building of a predictive model of the full-length APE1/DNA complex. Furthermore, the spatial configuration of the N-terminal tail is sensitive to NPM1, which could be related to the regulation of APE1.

## 1. Introduction

APE1 (apurinic apyrimidinic endonuclease 1) is a key enzyme of one of the major DNA repair routes, the BER (base excision repair) pathway [1,2,3,4]. The BER machinery is in charge of repairing some of the most frequent types of DNA damage, such as oxidative and alkylative, non-bulky base lesions [4,5,6]. Damaged nitrogenous bases are first detected and removed by lesion-specific DNA glycosylases, usually resulting in an abasic (apurinic/apyrimidinic, AP) site. Potentially mutagenic AP sites can also be generated through spontaneous or damage-induced depurination, such as that induced by reactive oxygen species (ROS), more frequent than depyrimidation. AP sites are recognised by APE1, which catalyses the incision of the phosphodiester bond in their 5′ side, leaving a break with 3′-hydroxyl and 5′-deoxyribose phosphate (5′-dRP) termini. After DNA strand incision by APE1, downstream BER enzymes complete the repair process. In the simplest, “short-patch” BER subpathway, DNA polymerase β (Pol β) uses its lyase domain to excise the 5′-dRP group left behind by APE1 and adds the corresponding filling nucleotide. Finally, the nick is then sealed by DNA ligase III. All of these APE1-downstream reactions are coordinated by the scaffolding protein XRCC1 (X-ray repair cross-complementing protein 1) [5,6]. In an alternative BER subpathway, termed “long-patch” (LP-BER), 2–10 nucleotides are replaced with the involvement of other polymerases (δ, ε) and ligase I, the sliding clamp PCNA acting as a binding platform [5].

In addition to the endonuclease activity, APE1 fulfils additional roles in the context of DNA repair, functioning also as an exonuclease. Thus, it exhibits 3′ end processing activities, removing terminal blocking groups in BER intermediates, e.g., rendered by some “bi-functional” glycosylases able to also cut the DNA strand, and proofreading DNA mismatches introduced by Pol β [1]. Furthermore, it participates in a DNA repair pathway known as nucleotide incision repair (NIR), by cutting non-abasic DNA at certain base lesions, and additionally contributes to the repair of single-strand breaks (SSBs) [1,7]. Besides its multiple DNA repair roles, APE1 acts as a redox factor regulating the DNA binding of several transcription factors, thus receiving the alternative name REF-1 [8].

Human APE1 (318 amino acids long) consists of a globular domain, where the nuclease activity resides, preceded by an N-terminal, ca. 40 residues long, intrinsically disordered region. While the globular domain of APE1 is sufficient for recognition and incision of abasic DNA [9], the flexible N-terminal domain, absent in prokaryotic homologues of APE1 [10], displays a regulatory role, being involved in mediating APE1 cell localisation, post-translational modifications, and protein–protein interactions [3]. Thus, the N-terminal tail is thought to be important for BER coordination [11,12], scanning of the DNA in search of lesions [10] and/or the RNA-related functions of APE1 [13]. Furthermore, although dispensable for incision activity, the N-terminal tail affects DNA binding and nuclease properties of APE1 [14], suggesting that it contributes somehow to the recognition and catalysis events. The functional roles of this region, however, remain poorly understood.

The 3D structure of the catalytic domain of APE1, either alone [15] or in a complex with different substrate and product-mimicking oligonucleotides [16,17,18] (Figure 1) has been deeply characterised. Of note, the 3D structure of the globular nuclease domain does not significantly change when APE1 binds to DNA [16], regardless of the DNA molecule being the abasic substrate or product. Based on the plentiful crystallographic and biochemical information available, the molecular details of APE1 catalytic activity have been untangled. However, in the APE1 structural models, the segment spanning at least 37 N-terminal residues is either absent in the APE1-expressing construct used [15,17] or not visible in the structure due to its flexibility [16].

Nevertheless, the N-terminal tail, rich in lysine residues especially in its N-terminal half, is probably involved in the interaction with DNA and/or RNA. Indeed, protection of some of these residues towards hydrogen/deuterium exchange has been observed upon DNA binding, and interestingly, they become less protected once the substrate is incised [19], suggesting that the contact of this region with the substrate DNA is not equivalent to that with the product. Moreover, in the context of BER coordination, the N-terminal tail might help APE1 remain attached to the incised product until release is permitted by the subsequent enzymes of the pathway. APE1 catalytic turnover is facilitated by Pol β and XRCC1, the latter serving as a binding platform, within the BER machinery [12]. The interaction of APE1 with nucleophosmin (NPM1) [14,20,21] has been recently proposed to have a similar role in the APE1 catalytic cycle [3]. NPM1, an abundant nucleolar protein, regulates multiple processes in cell homeostasis, such as ribosome biogenesis and centrosome duplication. It also stabilises some tumour suppressors (e.g., p53 and Arf), acts as a chaperone during nucleolar stress and participates in different DNA damage response (DDR) pathways [21]. NPM1 is an oligomeric, multi-domain protein of 294 residues, consisting of a pentameric core connected, through very long, flexible linkers, to small, globular C-terminal domains. The intrinsically disordered linkers make NPM1 prone to engage in phase separation phenomena, which has been related to APE1 regulation [22]. Furthermore, the N-terminal tail of APE1, also flexible, serves as a binding platform for several ligands including NPM1 [14,20,21] and XRCC1 [11] and could mediate the modulation of APE1 function through these proteins. However, the details of these regulatory mechanisms remain to be elucidated.

The flexible character of the N-terminal tail of APE1 hampers elucidation of its structural features. As mentioned, although the N-terminal tail is probably involved in the interaction, the binding of APE1 to nucleic acids is not enough to fix the structure of this region and solve its position in crystals made with the full-length protein [16]. This suggests that, even in the complexes with DNA, the N-terminal tail is mobile and establishes only transient interactions that can change along the catalysis. Thus, the mechanisms involved in APE1 regulation probably imply dynamic variations in the relative positions of the N-terminal tail with respect to the globular domain and of both domains with respect to the DNA. Indirect methods, such as spectroscopy, may provide information on the structural role of this region. Herein, we aim to explore the spatial configuration of the N-terminal region of APE1 when bound to abasic DNA or to NPM1, using Förster resonance energy transfer (FRET), a fluorescence spectroscopy technique sensitive to the distances between a fluorophore and a neighbouring acceptor. We have detected FRET-range communication between the two domains of APE1 and between them and oligonucleotide models of abasic DNA, observing intra- and intermolecular conformational rearrangements that may underpin APE1 function and regulation.

## 2. Results

### 2.1. APE1 Can Be Fluorescently Labelled with Cy3 and Cy5, Mostly Preserving Its Structure and Function

In order to analyse the role of the N-terminal tail of APE1 in the interaction with abasic DNA and NPM1, we labelled the recombinant protein with the fluorescent probes Cy3 and Cy5. They acted as FRET acceptors from fluorescein-labelled and Cy3-labelled oligonucleotides, respectively. APE1, as well as the truncated mutant ∆N33APE1 lacking the first N-terminal 33 residues, were labelled with Cy3 NHS-ester in primary amino groups, using conditions (i.e., pH 7.0 [23], see Materials and Methods Section) to favour reaction with the N-terminus only. An average degree of labelling (DOL) of 0.9 and 1.7 mol of dye per mol of protein was achieved for Cy3-APE1 and Cy3-∆N33APE1, respectively, indicating that some Lys residue may also be unintentionally labelled in the second case. In addition, we labelled full-length APE1 with Cy5 maleimide in thiol groups, obtaining a DOL of 1.1. APE1 contains 7 cysteines, most of which are buried within the protein and only two are partly exposed: C99 and C138, with an estimated solvent accessibility of 5–10% and more than 30%, respectively, of their total surface, based on the program SPDBviewer [24]. Thus, we assume that cysteine 138 is the main labelled residue when using Cy5 maleimide (Figure 1). We have also labelled Cy5-APE1 with Cy3 NHS-ester obtaining a doubly labelled protein (DL-APE1) with average DOL of 4.3 and 1.1 for Cy3 and Cy5, respectively. When some reactions with Cy3 NHS-ester rendered DOL values higher than 1.0, as in this case, we assume that most of the labelling targets the N-terminal region, harbouring 8 lysines in the first 33 residues. An analysis of all protein batches under study by SDS-PAGE is shown in Appendix A.

To evaluate the putative effect(s) of APE1 labelling on the protein structure and stability, we used circular dichroism (CD). There was no significant change in the far-UV spectral shape of APE1 upon labelling with Cy3, Cy5 or both, suggesting that its structure is preserved (Appendix A). Furthermore, all of the labelled samples showed a cooperative thermal unfolding with midpoints of 50–53 °C, as compared to wild-type, unlabelled APE1 with a described Tm of 52.2 °C in the presence of magnesium [20] (Appendix A). These results indicated the absence of significant effects of the labelling modifications on APE1 stability.

Recombinant APE1 binds with high-affinity dumbbell-shaped oligonucleotides containing tetrahydrofurane (THF), which are used as models of abasic substrate (“S”) and product (“P”) [17,20]. The DNA binding ability of the different labelled forms as compared to unlabelled APE1 was checked by native gel electrophoresis. We confirmed that the labelled batches were able to bind the model oligonucleotides, although binding is significantly weaker for the doubly labelled protein (Appendix A). Labelled APE1 seems to bind better to the substrate than the product (compare the intensity of the protein bands in Appendix A), yet another sign of specific recognition. Moreover, the binding affinity of Cy3-APE1 for Cy5-labelled product DNA has been quantified based on FRET signal saturation, rendering a K_D_ of 5 nM, almost identical to that of unlabelled APE1 (J.C. Penedo, personal communication). In addition, binding to DNA was described to thermally stabilise APE1 [20], and the stabilising effect could also be observed for Cy3 and Cy5-APE1, although not for the doubly labelled protein, in agreement with its apparent weaker affinity for the DNA (Appendix A). Finally, we performed incision assays followed by polyacrylamide-urea denaturing electrophoresis. Because of the closed, dumbbell shape of the substrate oligo, it is not split by the incision (Appendix A), still, the closed substrate and open product migrate differently in these gels. We have found that all of the different samples of labelled APE1 are indeed active in hydrolysing the oligonucleotide model of abasic substrate, although the activity of Cy5-APE1 is lower and, as expected, that of DL-APE1 is almost negligible as compared with unlabelled APE1 (Appendix A).

Altogether, these control experiments indicated that APE1 conformation and function are largely preserved upon labelling with Cy3 or Cy5 dyes and that the labelled protein is a valid model to explore the interaction of APE1 with DNA.

### 2.2. FRET between Models of Abasic DNA and APE1 Serves as Probe of the Relative Location of the N-Terminal Tail

In order to explore the role of the N-terminal domain of APE1 in the interaction with abasic DNA, we have measured the FRET efficiency between a tetrahydrofurane (THF)-containing, dumbbell-shaped oligonucleotide, model of abasic product (“P”) [20] and recombinant APE1, either full length or the truncated mutant ∆N33APE1. The oligonucleotide was fluorescently labelled with fluorescein or Cy3 in either of two residues: thymine 9 (in the same strand, 8 nucleotides away from the abasic site on the 3′ side) or thymine 29 (in the opposite strand, 5 nucleotides away in the 5′ side) (Appendix A). Additionally, APE1 was labelled with Cy3 in the N-terminal region (mostly N-terminus) or with Cy5 in cysteines (mostly C138), sited in the globular domain (Figure 1). Thus, the putative transfer between fluorescein and Cy3 or between Cy3 and Cy5, both of them being well-established FRET pairs [25] has been evaluated.

The mixture of oligonucleotide “P” labelled with Cy3 (Cy3-P) and Cy5-APE1 in equimolar amounts shows a decrease in Cy3 and an increase in Cy5 emission intensities as compared with the controls with unlabelled protein or unlabelled DNA, indicating that FRET between the two dyes takes place (Figure 2). When Cy3 is on thymine 29 of the oligo (T29-Cy3-P), the FRET efficiency is higher (25.7 ± 8.0%) than when the label is on thymine 9 (T9-Cy3-P, 14.6 ± 3.5%) (Figure 2 and Figure 3). These data are calculated based on the “acceptor sensitisation method”, which corrects for any artefactual contribution of fluorescence quenching or enhancement upon binding [25]. As discussed above, the DOL of 1.1 in Cy5-APE1 and the presence of only one significantly exposed cysteine, C138, allow assuming that it is that residue the one labelled. On this basis, our observed FRET efficiencies are in agreement with the closer distance from cysteine 138 of APE1 to T29, i.e., upstream of the abasic site, than to T9, downstream, in the 3D structure of the complex APE1/product DNA (PDB entry code 5DFF [17]) (Figure 1). The FRET efficiency depends on the sixth power of the distance between donor and acceptor, and on the R_0_ parameter (53 Å for the Cy3-Cy5 pair) according to Equation (1). However, in practice, the transfer efficiency is affected by other factors such as the relative orientation of the dyes [26]. Furthermore, the distance model is not exactly applicable in a system with multiple donors and acceptors, i.e., if the DOL values are not exactly 1.0 [27] and therefore, the estimation of distances from FRET efficiencies is very uncertain. Thus, the application of Equation (1) to our measured FRET efficiencies renders relative estimated distances of 63.7 Å for T29-Cy3-P/Cy5-APE1 and 71.4 Å for T9-Cy3-P/Cy5-APE1, while the distances between the C138 (S) of APE1 and the C5 of those two nucleotides in the 3D structure of the complex (5DFF) [17] are 36.2 and 63.6 Å, respectively. One should consider that the labelling linkers may contribute up to an additional 15–20 Å to the effective distance between donor and acceptor, and that mobility of the N-terminus may also account for an overestimation of its distance to DNA.

We have also evaluated FRET between the oligonucleotide P labelled with fluorescein on T9 or T29 (T9-fluor-P or T29-fluor-P, respectively) and N-terminally labelled Cy3-APE1. In this setting, the measured efficiency for the T9-fluor-P/Cy3-APE1 pair was higher (19.5 ± 5.9%) than for T29-fluor-P/Cy3-APE1 (7.0 ± 2.7%) (Figure 3), indicating that the N-terminus of the protein locates closer to T9 than to T29. This observation is not surprising given that the first residue visible in the structure [17], connecting to the N-terminal region, also faces towards T9. The efficiency and thus the distance between T9 and the N-terminus seems to be quite similar to T29/C138 (see above), suggesting that the long, flexible, N-terminal region does not extend far away but remains relatively close to the DNA.

We also wanted to probe an additional position within the N-terminal tail of APE1. To that aim, we labelled the N-terminus of the truncated mutant ∆N33APE1 with Cy3 and tested the FRET from T9-fluor-P and T29-fluor-P. The observed efficiencies were significantly lower than for the full-length protein (Figure 3), which would mean that residue 34 is located further apart from the DNA than the N-terminus. Thus, the FRET data suggest that the N-terminus locates close to the DNA bases and then the polypeptide chain runs further away before connecting back to the globular domain. As in the case of the full-length protein, the transfer to an acceptor in the N-terminus of ∆N33APE1 is higher from T9 than from T29 (where it is not detectable), reinforcing the idea that the entire N-terminal region is located at the downstream side of the abasic site. Based on these findings, and taking the estimated distances only as relative dimensions, we have attempted to model the approximate position of the N-terminal tail within the 3D structure of the APE1/DNA complex (Figure 4). For this model, we have first run a prediction with AlphaFold [28] (Appendix A) and then modified the position of the N-terminal region according to the FRET-based relative distances. One must note nevertheless that the reliability of AlphaFold prediction is “low” or “very low” for the most part of the N-terminal tail (per-residue confidence score between 31.93 and 52.69 for residues 1–40) (Appendix A), as expected for intrinsically disordered regions [29].

Finally, we hypothesised that the N-terminal tail of APE1 changes its configuration upon catalytic incision. To explore this possibility, we have analysed whether the FRET between the DNA and APE1 is sensitive to the catalytic state of the DNA, i.e., whether the employed oligonucleotide is open at the 5′ side of THF, corresponding to a model of the abasic product (P) or else, is a closed dumbbell, resembling the substrate (S). We have compared the FRET efficiency from T9-fluorescein labelled oligonucleotide models of substrate or product (T9-fluor-S and T9-fluor-P) to Cy3-APE1. We observed only a small difference (17.2 vs. 19.6% for the substrate and product, respectively) (Figure 3), suggesting that the N-terminus locates slightly further from the DNA in the case of the substrate, but further analysis would be required to confirm this observation.

### 2.3. Intramolecular FRET Reflects Proximity between Both Domains of APE1, and Is Sensitive to NPM1 Binding

To explore the relative position of the N-terminal tail with respect to the globular domain of APE1, and how it is affected by interaction with DNA or NPM1, we have doubly labelled full-length recombinant APE1 with Cy3, mostly in the N-terminal region, and Cy5 in thiol groups, mostly C138. The emission spectrum of this doubly labelled protein (“DL-APE1”) when excited at the donor λ_exc_ (547 nm) is shown in Figure 5A. The Cy5 emission intensity in this sample is much higher than Cy5-only labelled APE1 with the same degree of Cy5 labelling (Appendix A). Although in the absence of the “donor control”, determination of absolute efficiencies was not reliable, the comparison clearly indicates a significant FRET phenomenon in DL-APE1. Furthermore, the Cy5 emission intensity is more than double that of the donor (Cy3), whereas in denaturing conditions (6 M guanidinium hydrochloride) the relative Cy5 emission is lower than that of Cy3 (Figure 5), indicating that the transfer is abolished to a large extent when the protein is unfolded; in other words, the protein N-terminus and the globular domain are much closer in the native structure than in the unfolded state.

The presence of the oligonucleotides P or S in ten times molar excess does not induce significant changes in the acceptor/donor intensity ratio (A/D) (Figure 5B), which could mean that DNA binding does not imply a gross variation in the relative position of the N-terminal tail with respect to the globular domain of APE1. Nevertheless, the weaker affinity for DNA displayed by DL-APE1 (see above) precludes a clear interpretation of this result. By contrast, when mixing DL-APE1 with NPM1 in a 1:10 APE1:NPM1 pentamer molar ratio, either with or without DNA, the A/D ratio increases by a factor of 20% (Figure 5B), suggesting that interaction with NPM1, which is mainly mediated by APE1 N-terminal tail [14,20], induces a conformational rearrangement in the protein, where this region becomes closer to C138 in the globular domain.

### 2.4. Effect of NPM1 on APE1/DNA Interaction as Probed by FRET

NPM1, which binds to the APE1 N-terminal region, has been proposed to regulate the BER machinery by favouring APE1 specific recognition of the abasic substrate and dissociation from the incised product thus facilitating APE1 turnover [20]. To analyse whether the FRET between DNA and APE1 is sensitive to the presence of NPM1, we have recorded fluorescence emission spectra of labelled DNA/APE1 pairs in the presence of an excess of NPM1. These experiments were performed only with full-length APE1 since the truncated form hardly binds NPM1. We have obtained lower transfer efficiencies (not statistically significant differences) in ternary mixtures of Cy3-labelled DNA/Cy5-labelled APE1/NPM1 than in the absence of NPM1 (Figure 3). This could be explained by a partial dissociation of the protein from the DNA or by a conformational rearrangement implying a longer distance between DNA and the globular domain of APE1. By contrast, FRET between fluor-DNA and Cy3-APE1 is more efficient in the presence of NPM1, and the difference is statistically significant for T9-fluor-P (31.5 vs. 19.6%) (Figure 3 and Figure 6), meaning that NPM1 binding induces a further approaching of the N-terminus to the DNA, especially to T9, i.e., the 3′ side of the abasic site. Interestingly, this effect of NPM1 is also observed with the substrate DNA (24.3 vs. 17.2%), although the relative positions of the N-terminus in both presence and absence of NPM1 would be further apart from the DNA as compared to the product DNA (Figure 3). Moreover, the fact that the NPM1-induced effect is not equally sensed when the DNA label is on T9 or on T29 (Figure 3) further supports the idea that a specific conformational change takes place in the APE1/DNA complex upon NPM1 binding to APE1. Of note, the NPM1-induced approaching of the N-terminus to T9 also implies it coming closer to the globular domain of APE1, in agreement with the effect of NPM1 on the doubly labelled APE1 described above. Altogether, our FRET data indicate that in a complex with abasic DNA, the N-terminus of APE1 approaches the double helix at the 3′ side of the abasic site, its spatial configuration being sensitive to the DNA catalytic state and to the presence of the regulatory protein NPM1.

## 3. Discussion

APE1, the essential endonuclease in the BER pathway of DNA repair, consists of a catalytic globular domain preceded by a flexible N-terminal region, which displays a regulatory role [1,2,3]. As for other flexible protein regions hardly susceptible to structural elucidation, the molecular mechanisms governing the modulation of APE1 activity mediated by the N-terminal tail are far from being understood. While not essential for endonuclease catalysis and not defined or visible in crystal structures of APE1/DNA complexes, the N-terminal tail, rich in positively charged residues, probably takes part in APE1 interaction with nucleic acids [3,19]. Moreover, as a platform for protein–protein interactions, the N-terminal tail could mediate the facilitation of enzymatic turnover brought about by several APE1 ligands, as suggested for XRCC1 [11] and NPM1 [3]. Indeed, increasing evidence points to a role for NPM1 in the DNA damage response (DDR) [30] and in BER in particular [21], but the underlying mechanisms of this function remain to be described.

Herein, we have explored the role of the APE1 N-terminal tail in the interaction of the protein with oligonucleotides mimicking abasic DNA, using intensity-based FRET assays. The non-radiative transfer of energy from a fluorophore to neighbouring molecules makes the FRET phenomenon sensitive to molecular distances. Thus, it serves as a “ruler” with a resolution adequate for biological complexes and is particularly suited for dynamic systems, not amenable to conventional structural techniques [25]. We have detected FRET with the fluorescein/Cy3 and Cy3/Cy5 pairs, labelling the THF-containing oligonucleotide and APE1 in different positions. By comparatively quantifying variations in FRET efficiency, we have observed changes in relative positions of the N-terminal tail with respect to the globular domain of APE1 and of them with respect to DNA duplexes in the substrate and product configuration. According to our data, the N-terminus is closer to the DNA than residue 34, and both residues locate closer to the 3′ side than to the 5′ side of the abasic site. Thus, the N-terminus itself seems to have a more prominent role in the interaction with the DNA, while the rest of the N-terminal tail may remain mobile. Our data additionally suggest that the N-terminus is closer to the DNA in the post-incision state than to the substrate, whereby it might have a role in maintaining APE1 bound to the product until the arrival of the next enzyme of the route. This coordinated action of the BER enzymes has been termed “passing the baton” [31].

NPM1, a nuclear chaperone, has been reported to bind the N-terminal tail of APE1 and regulate its functionality [14,20]. Our data indicate that NPM1 increases the intramolecular FRET between the N-terminal region and cysteine 138 of APE1, suggesting that a conformational rearrangement is induced in APE1 as a result of NPM1 binding, wherein the N-terminus gets closer to the globular domain. Furthermore, in the context of the APE1/DNA complex, NPM1 seems to increase the proximity between the N-terminus and the DNA product or substrate. Notably, this movement also implies that the N terminus approaches the protein globular domain, in agreement with our intramolecular FRET data using a doubly labelled APE1. These findings suggest that NPM1 might help keep APE1 bound to the product, in a relatively closed state, until it is handed to the next enzyme, Pol β. Eventually, once turnover is permitted, it might also favour detachment of the protein from its already incised product. This facilitating role has been proposed for NPM1 [3] and other regulatory proteins [11]. The conformational rearrangements we describe suggest a possible mechanism: The distance between the highly basic N-terminus and a positively charged patch in the DNA-binding site of APE1, including, e.g., Arg177, Lys 224 and Lys 228 (Appendix A) would be shortened upon incision and especially upon NPM1 binding. This could cause electrostatic repulsion, thus facilitating the detachment of the globular domain, and subsequently of the entire protein, from the already incised DNA.

In summary, using a combination of intra- and intermolecular FRET assays, we have explored the relative positioning of the N-terminal tail that cannot be resolved in crystal structures of APE1, either in the apo state or when bound to DNA duplexes mimicking the abasic substrate or product. Moreover, we have been able to detect conformational changes involving the N-terminal region of APE1, finding that NPM1 induces closure of the protein around the DNA, which, through an electrostatically modulated mechanism, might promote the enzyme dissociation from the product DNA.

## 4. Materials and Methods

### 4.1. Oligonucleotides

The fluorescein or Cy3-labelled (or unlabelled, as control) oligonucleotides were synthesised by IDT (Leuven, Belgium). As a model of abasic DNA product of APE1, we used an oligonucleotide containing tetrahydrofurane (THF), an abasic residue analogue [17]: 5′-XCGGTCGATCGTAAGATCGACCGTGCGCTGGAGCTTGCTCCAGCGC. This sequence is expected to adopt a dumbbell-shape with the THF centrally located. The fluorescein or Cy3 were attached to thymines 9 or 29 (underlined) (see Appendix A). The oligos were resuspended at 100 μM in water, annealed by heating 5 min at 95 °C and slowly cooling down in a thermoblock, and stored at −20 °C. To prepare the “substrate”, the same sequence but permuted, not starting with X, was used: 5′-CTGGAGCTT GCTCCAGCGCXCGGTCGATCGTAAGATCGACCGTGCG-3′, and ligated with T4 ligase (New England Biolabs, Ipswich, MA, USA) (4000 U ligase per nmol of DNA).

### 4.2. Protein Production and Labelling

APE1 (full length and APE1∆N33) and NPM1 were overexpressed in *E. coli* as N-terminally His-ZZ-tagged constructs and purified as described previously [20,32]. Briefly, purification involved Ni-NTA chromatography, tags removal with TEV protease, reverse Ni-NTA and a final size exclusion chromatography (SEC) step. APE1 (full length and APE1∆N33) was labelled in primary amino groups with NHS-Cy3 (Jena Bioscience, Jena, Germany) in 20 mM potassium phosphate pH 7.0, 50 mM NaCl (labelling buffer). Note that at pH 7.0, the N-terminal amino group is selectively labelled [23]. The protein, at approx. 5 mg/mL, was mixed with a ten times smaller volume of NHS-Cy3 dissolved at 2.5 mg/mL in dimethylformamide. The mixture was incubated for 1 h at room temperature and mildly shaken every 15 min. The labelled protein was then separated from excess dye by means of a PD-10 column, and its concentration and degree of labelling (DOL) were estimated based on the comparison of absorbances at 280 and 550 nm. Labelling with Cy5 on cysteines was performed with Amersham Cy5 maleimide mono-reacive dye (GE Healthcare). APE1 at ca. 3 mg/mL in labelling buffer was mixed with a 5% volume of the Cy5 reagent, incubated during 2 h at room temperature and separated from free dye with PD-10. Determination of protein concentration and DOL was performed by measuring the absorbances at 280 and 650 nm. For double labelling of APE1, the protein was first labelled with Cy5 and then with Cy3.

### 4.3. FRET Experiments, Fluorescence Spectroscopy and Analysis

Mixtures of 500 nM donor-labelled oligonucleotide and 500 nM acceptor-labelled APE1 (either full-length or APE1∆N33) were prepared in 300 μL of 20 mM potassium phosphate pH 7.0, 50 mM NaCl, 2 mM DTT, 10 mM EDTA (FRET buffer) and incubated for 30 min at room temperature. Control mixtures containing only the donor or acceptor fluorophores were prepared with unlabelled protein or unlabelled DNA, respectively. In the experiments with NPM1, ternary mixtures included 4 μM NPM1 pentamer (unlabelled). Fluorescence emission spectra were recorded in a Fluoromax-3 fluorimeter (Jobin Yvon Horiba, Kioto, Japan) with slits width of 2 nm, at 20 °C. In the case of the fluorescein/Cy3 FRET pair, exciting was at 494 nm, except for the “acceptor 100% control”, where λ_exc_ was 547 nm. For the Cy3/Cy5 pair, λ_exc_ was 547 nm, or 647 nm for the 100% control. In the intramolecular FRET experiments, 50 nM doubly labelled (DL)-APE1 in FRET buffer was measured alone or after incubation for 30 min with 500 nM DNA (either product or substrate), with 500 nM NPM1 pentamer, or mixtures thereof, or in the presence of 6 M guanidinium hydrochloride. Emission spectra were recorded in the range 552–600 nm upon excitation at 547 nm, using 4 nm-width slits.

FRET efficiency E was determined based on the enhancement of acceptor emission, according to Blouin et al., 2015 [25]. The “FRET spectrum” of the mixture, AD, was first normalised to the “donor” control, D. Then, the fluorescence intensity (I_f_) of the acceptor (at 565 nm for Cy3, at 665 nm for Cy5) in the FRET spectrum was corrected by subtracting the I_f_ of the acceptor control, A, and this value was then compared to the I_f_ of the “100% control” (see Figure 2). Donor–acceptor distances (R) were estimated from [25]:E = R_0_^6^/(R_0_^6^ + R^6^)(1)
using R_0_ values of 56 Å and 53 Å for the fluorescein/Cy3 and Cy3/Cy5 pairs, respectively.

### 4.4. Circular Dichroism

Circular dichroism measurements were performed in a Jasco 720 spectropolarimeter (Tokyo, Japan) equipped with Peltier temperature control, in a cuvette of 0.2 cm pathlength, in buffer 20 mM potassium phosphate pH 7.0, 50 mM NaCl, 5 mM MgCl_2_. Thermal scans based on ellipticity at 222 nm were performed at a heating rate of 1 °C/min. Protein and oligo concentration was 4 μM.

### 4.5. APE1 Binding Assays

APE1/DNA binding was assayed through native electrophoresis [20]. The different APE1 variants (unlabelled, Cy3- Cy5- and DL) at 2 μM were mixed with an equimolar amount of the oligonucleotide (either product or substrate, and either labelled or unlabelled), in buffer 20 mM potassium phosphate pH 7.0, 50 mM NaCl, 5 mM MgCl_2_, 2 mM DTT, 0.01% (*w*/*v*) Tween-20 (total volume of 20 μL). After incubation for 30 min at room temperature, the samples were loaded in precast native 4–16% polyacrylamide (PA), 10 wells Bis-Tris gels (Invitrogen, Waltham, MA, USA). Running buffer and sample buffer (without G-250) were also from Invitrogen. Gels were run for 130 min at 150 V and 4 °C, first visualised without staining, detecting the dye’s fluorescence, then stained with GelRed (Biotium, Fermont, CA, USA) and then with Coomassie, and photographed with Gel Doc EZ System (Bio-Rad, Hercules, CA, USA). Densitometry analysis of the protein bands was performed with Quantity One (Bio-Rad).

### 4.6. APE1 Incision Assays

Incision of 2 μM abasic substrate model oligonucleotide was assayed at 37 °C in buffer 20 mM potassium phosphate, 50 mM NaCl, 5 mM MgCl_2_, 2 mM DTT, pH 7.0, and monitored by denaturing 18% polyacrylamide (PA)-urea gels [20]. The samples were incubated during 0–30 min at 37 °C in the presence of 5 nM (unlabelled, Cy3-, Cy5-APE1) or 500 nM (DL-APE1) enzyme. Reactions were halted by the addition of the same volume of loading solution containing 96% formamide, 10 mM EDTA and 0.1% bromophenol blue, and heating the sample at 95 °C for 5 min, prior to electrophoresis. “Product” oligonucleotide was also loaded as a reference. Ten well, 1 mm thick, 18% PA gels containing 8 M urea were pre-run for 30 min, loaded with 20 μL of sample per well, and run for 140 min at 175 V. Gels were stained with GelRed and analysed for densitometry with Gel Doc EZ and Quantity One (Bio-Rad).

### 4.7. Protein Modelling

All analysis and modelling were based on the 3D structure 5DFF of the globular domain of APE1 bound to a model of abasic product [17]. Structures were displayed with PyMOL (The PyMOL Molecular Graphics System, Version 2.0 Schrödinger, LLC).

## Figures and Tables

**Figure 1 ijms-23-08015-f001:**
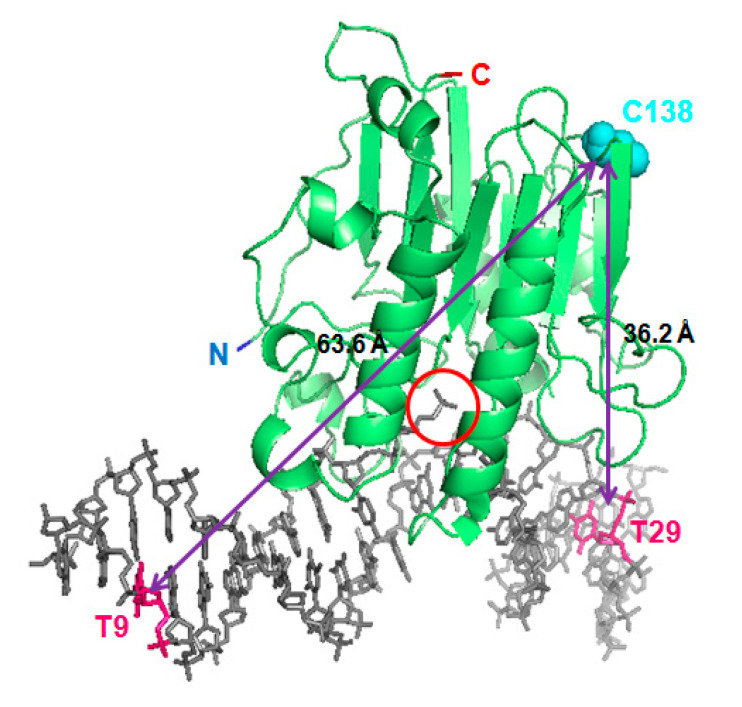
Fluorescent labelling positions shown on the APE1/product complex. The 3D structure of APE1 (starting residue 43) bound to an oligonucleotide mimicking the abasic product (5DFF [17]) is shown as cartoon (protein) and sticks (DNA) representation. Labelled residues in the protein (cysteine 138, in cyan) and in the DNA (thymines 9 and 29, in magenta) are highlighted. The open abasic site is encircled, and the protein termini are also indicated.

**Figure 2 ijms-23-08015-f002:**
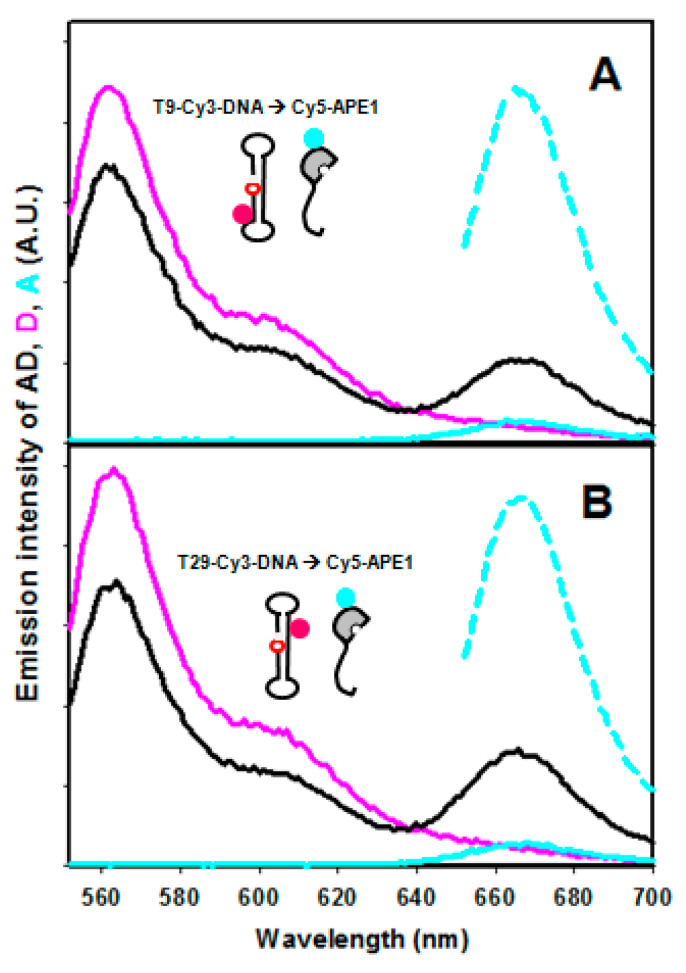
FRET spectra. Fluorescence emission spectrum of a mixture of Cy3-DNA and Cy5-APE1 at 1:1 molar ratio (donor-acceptor pair, DA, black line) upon excitation at 547 nm, as compared with controls with unlabelled acceptor (D, magenta) or unlabelled donor (A, cyan, solid line). The acceptor in the presence of unlabelled donor was also excited at 647 nm (“100% control”, cyan broken line). The labelling of the DNA was on T9 (**A**) or T29 (**B**).

**Figure 3 ijms-23-08015-f003:**
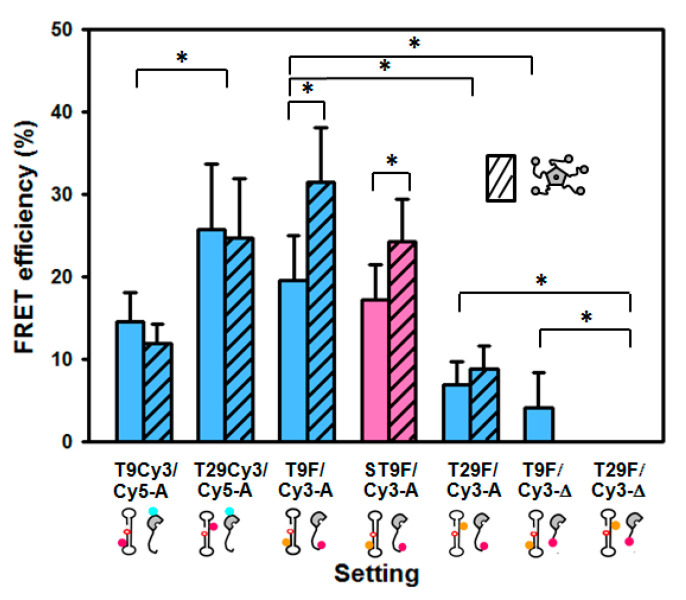
FRET efficiencies for the different APE1/DNA settings, in the absence (empty bars) or presence (striped bars) of NPM1. The values are averages of 3–7 experiments, with the standard deviation. All of the DNAs are models of APE1 product, except for the one shown in pink, corresponding to T9-fluor-DNA substrate. Asterisks indicate statistical significance (*p* < 0.05).

**Figure 4 ijms-23-08015-f004:**
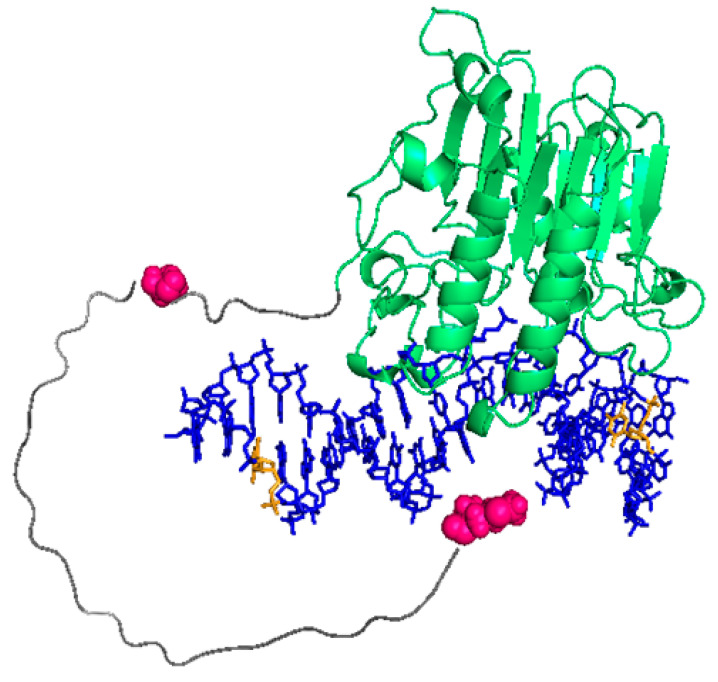
Predictive model of APE1 N-terminal region within the APE1/DNA complex. The 42 aa-long N-terminal region (grey) was firstly modelled with AlphaFold and positioned onto the APE1/DNA complex (5DFF [17]), as based on relative distances from FRET results. APE1 residues 1 and 34 (pink) and nucleotides T9 and T29 (yellow) are highlighted.

**Figure 5 ijms-23-08015-f005:**
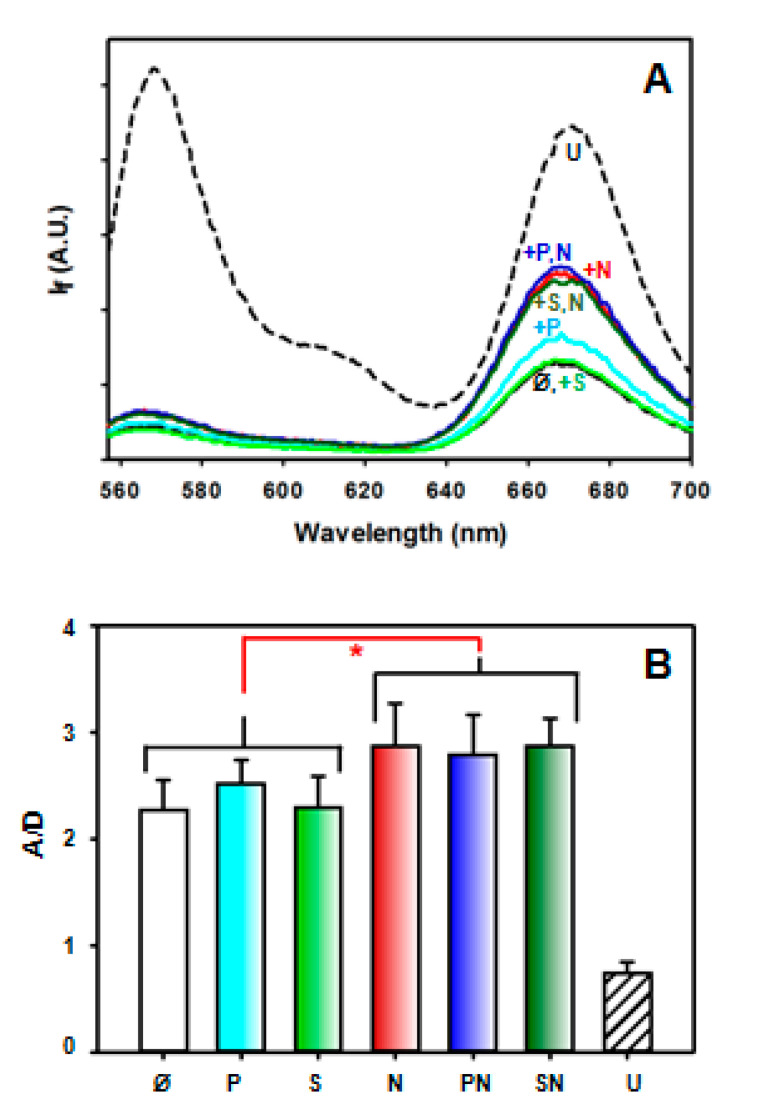
(**A**) Spectra of 50 nM doubly labelled (DL) APE1 alone and in the presence of DNA substrate (S), product (P), or NPM1 (N), 500 nM each, or mixtures thereof. Note that the spectra of the protein alone (solid black line) and with substrate (green) appear overlapped. The spectrum of the unfolded protein (U, broken line) is also shown. (**B**) Ratios of the emission intensity at 665 nm (Cy5, acceptor) divided by that of Cy3 (donor) at 565 nm. The values are averages of 4 measurements. The difference between the three groups without (∅, P, S) or with NPM1 (N, PN, SN) Asterisks is statistically significant (*p* < 0.05).

**Figure 6 ijms-23-08015-f006:**
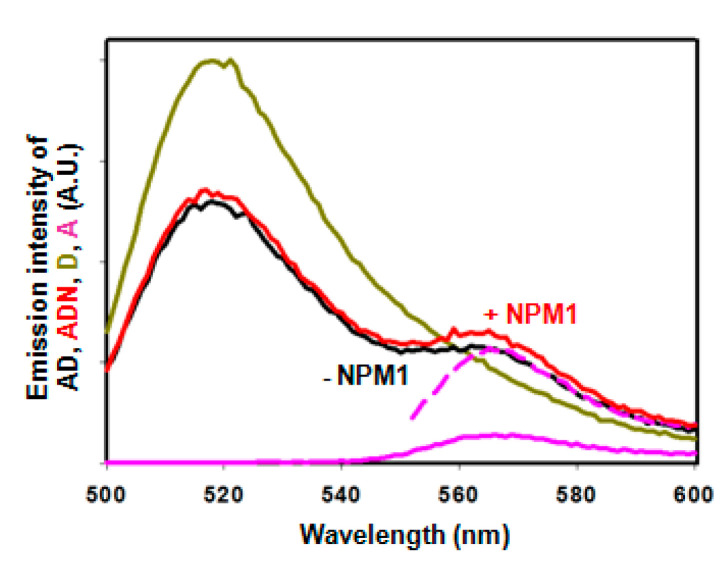
FRET spectrum of the mixture T9-fluor-DNA product/Cy3-APE1 in the presence (red) and absence (black) of 4 μM NPM1 pentamer, together with the donor (dark yellow), acceptor (magenta) and “100% acceptor” (magenta, broken line) controls.

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
