# Peer review of "Conformational Rearrangements Regulating the DNA Repair Protein APE1"

_ijms, 2022, doi:10.3390/ijms23148015_

Round 1

Reviewer 1 Report

APE1 incises the abasic sites generated as intermediates during base excision repair pathway.  The N-terminal region of the human APE1 has roles in interaction with substrate DNA and other proteins.  The N-terminal region of the human APE1 has not been structurally characterized and the dynamics of the region remains elusive.  In this paper, authors described the FRET-based analysis to reveal the conformational rearrangement of the N-terminal region upon binding of DNA and NPM1.  The results of the FRET analyses indicate approach of the N-terminal region to the DNA or the globular domain upon interaction with product DNA or NPM1, respectively.  Based on these results, authors proposed a hypothesis that approach of the N-terminal basic region to the positively-charged DNA-binding site of APE1 causes an electrostatic repulsion, which facilitates turnover of the enzyme. Results described are clear and the proposed hypothesis is attractive. However, authors’ interpretations of the results of FRET analyses are overstated in several respects.

Major concerns

1.    The reviewer agrees that FRET is an appropriate approach to reveal the dynamics of the N-terminal region of the human APE1. The results are relatively clear, which simply indicates that the N-terminal region “approaches” the DNA or the globular domain upon binding of product DNA or NPM1.  However, authors claim that, based on the results of the FRET analyses, the N-terminal region “embraces” the DNA.  Authors make that claim probably because they trust the model structure of the full-length APE1 constructed by AlphaFold2.  The reviewer also constructed the model structure using AlphaFold2, and noted that the reliability is extremely low at the N-terminal region of the model. Accordingly, there is no result to support that the N-terminal region “embraces” the DNA.  

2.    As described above, the reliability of the AlphaFold2-based model structure is extremely low at the N-terminal region.  The model structure should be removed or colored according to the reliability.

Minor points

1.    The number of trials should be stated in the Fig. 3B.

Author Response

  1. The reviewer agrees that FRET is an appropriate approach to reveal the dynamics of the N-terminal region of the human APE1. The results are relatively clear, which simply indicates that the N-terminal region “approaches” the DNA or the globular domain upon binding of product DNA or NPM1. However, authors claim that, based on the results of the FRET analyses, the N-terminal region “embraces” the DNA.  Authors make that claim probably because they trust the model structure of the full-length APE1 constructed by AlphaFold2.  The reviewer also constructed the model structure using AlphaFold2, and noted that the reliability is extremely low at the N-terminal region of the model. Accordingly, there is no result to support that the N-terminal region “embraces” the DNA. 

We thank the reviewer for helping us to clarify this point. As mentioned by the reviewer, FRET is an appropriate approach, and the results provide clear evidence that the N-terminal region approaches the DNA/globular domain upon binding. To rationalize this experimental finding, we have explored potential orientations of the N-terminal region with respect to the rest of the complex using AlphaFold. We completely agree with the reviewer that, as expected for intrinsically disordered proteins (Ruff and Pappu, 2021, reference 29 in the revised version), the reliability of AlphaFold in this scenario is very low. Following the reviewer’s recommendation, we have removed the term ‘embrace’ and rewritten the corresponding parts in Results and Discussion sections to explicitly clarify the low reliability of the AlphaFold model for most of the segment and its use only as a predictive tool. As requested, we have also added a supplementary figure showing the reliability score of the model (Fig S6).

  1. As described above, the reliability of the AlphaFold2-based model structure is extremely low at the N-terminal region. The model structure should be removed or colored according to the reliability.

We have changed the image of the model (Fig. 4), representing the N-terminal region in a different color to clearly distinguish it from the crystallographic structure. The low confidence score of the prediction is thoroughly discussed in the text. In the new Fig. S6, the model is colored according to reliability.

 Minor points

  1. The number of trials should be stated in the Fig. 3B.

There is no Fig. 3B. In Fig. 3, we state the number of experiments (between 3 and 7) per condition. If you mean Fig. 5B, the number of trials was 4 and we have now mentioned it in the figure legend.

Reviewer 2 Report

In the manuscript of “Conformational rearrangements regulating the DNA repair protein APE1”, the author used FRET experiment shows APE1 N-terminal tail embraces the DNA at the downstream side of the abasic site and enables the building of a predictive model of the full-length APE1 / DNA complex. Furthermore, the spatial configuration of the N-terminal tail is sensitive to NPM1, which could be related to the regulation of APE1.

The comments:

1.       In Figure S1, why does APE1-WT and ∆N33-APE1 truncation show the same MW size, and please show the Marker together with the protein sample. 2.       Cartoon or sticks are two different modes to show the structure, so in Figure 1, DNA can only show in one mode like in sticks mode.

3.       The CD result is important to show that labeling does not affect the structure of APE1, even if it is the same, the author can put the result in supplementary data.

4.       From the DNA binding experiment, The DL protein shows the smear on the gel, so the thermal experiment is very important to show the stability of DL protein binding with DNA, however, the author used “not shown” data again. In addition, how to calculate the KD value of APE1 binding with DNA also need to be explained.

5.       In Figure 3, from the structure of APE1 it is already known that the N-ter of APE1 is close to T9 in the DNA. So the FRET efficiency of T9F/Cys3-A shows stronger than T29F/Cys3-A is normally.

6.       What does this sentence mean: “which would mean that residue 34, closer in sequence to the globular domain, is located further apart from the DNA than the N-terminus.” How does the author get this conclusion, especially residue 34? The FRET efficiency does not mean the 3D structure position of the N-ter of APE1, just speculate the N-ter should be close to DNA.

7.       Does the FRET efficiency of Cy3 show a difference between APE1 FL and ΔN33APE1 truncation? There are also many lysine residues in APE1 C-ter, like K303, K299, K276, K227, K228 …, from the structure these residues show much closer to T29 of the DNA, it is difficult to understand why that cy3 ΔN33APE1 /T29F shows no FRET efficiency.

8.    On page8 lines 287-289, how to calculate the increase of the factor of 20%?

9.       What is the novel finding of the Effects of NPM1 on APE1/DNA interaction from this manuscript to the References 19, which the authors published in 2020. 

                     The other part also needs to be improved:

1.       The introduction needs to be re-written please update the reference and organize it.

2.       The first paragraph of the introduction simply introduces the BER pathway, however, it needs to make more clear about the BER pathway, such as, in the short-patch BER pathway, how XRCC1 works with polymerase beta and Ligase III to finish the BER pathway. In addition, what is the long-patch BER pathway? Please update the references for this part.  DNA polymerase β is not DNA polymerase b.

3.       In the second paragraph, the endonuclease activity of APE1 is the activity of APE1 incision of the phosphodiester bond with the AP site. To remove the 3’-OH end is APE1 exonuclease activity. So please make this clearly.

4.       “Thanks to “… is not used in scientific reports.

5.       What is  “avidly tetrahydrofurane”?

6.       In the method part, there is no thermal experiment, the DNA labeling describe. 

Author Response

  1. In Figure S1, why does APE1-WT and ∆N33-APE1 truncation show the same MW size, and please show the Marker together with the protein sample.

Indeed, the lanes of full-length and ∆N33-APE1 come from different gels and they were wrongly aligned for the figure. We have now included the markers lane as internal reference of each gel, and thus it is evident that the two variants migrate differently, according to a size difference of 3421 Da.

  1. Cartoon or sticks are two different modes to show the structure, so in Figure 1, DNA can only show in one mode like in sticks mode.

This was also a mistake and has been corrected in the legend of Fig. 1

  1. The CD result is important to show that labeling does not affect the structure of APE1, even if it is the same, the author can put the result in supplementary data.

The CD spectra have been included in an additional panel of the supplementary figure, as well as the temperature scans of the protein variants in the presence of DNA (see below).

  1. From the DNA binding experiment, The DL protein shows the smear on the gel, so the thermal experiment is very important to show the stability of DL protein binding with DNA, however, the author used “not shown” data again. In addition, how to calculate the KD value of APE1 binding with DNA also need to be explained.

The thermal sscans in the presence of DNA have been included in Fig. S2.

In this case, the mentioned KD has been estimated from a binding curve based on the FRET efficiency upon addition of increasing amounts of Cy5-labelled DNA to Cy3-APE1, and this has been now mentioned in the text.

  1. In Figure 3, from the structure of APE1 it is already known that the N-ter of APE1 is close to T9 in the DNA. So the FRET efficiency of T9F/Cys3-A shows stronger than T29F/Cys3-A is normally.

We agree that the observation may be expected considering the crystal structure of the complex, and we have explained it in the text. Nevertheless, the N-terminal region, given its length, might reach the other side of the protein (facing T29) and our data indicate that this is not the case.

  1. What does this sentence mean: “which would mean that residue 34, closer in sequence to the globular domain, is located further apart from the DNA than the N-terminus.” How does the author get this conclusion, especially residue 34? The FRET efficiency does not mean the 3D structure position of the N-ter of APE1, just speculate the N-ter should be close to DNA.

We conclude this from the observation that FRET from the DNA to N-terminally labelled ∆N33-APE1 (i.e. corresponding to residue 34 of full-length APE1) is much less efficient than to N-terminally labelled full-length APE1.

Indeed, based on our data, we cannot claim to know the 3D position of any residue, we are just proposing a model compatible with the experimental data. Still, the entire description and discussion about the model has been rewritten in more hypothetical terms, following also the first reviewer´s suggestions.

  1. Does the FRET efficiency of Cy3 show a difference between APE1 FL and ΔN33APE1 truncation? There are also many lysine residues in APE1 C-ter, like K303, K299, K276, K227, K228 …, from the structure these residues show much closer to T29 of the DNA, it is difficult to understand why that cy3 ΔN33APE1 /T29F shows no FRET efficiency.

Yes, there are statistically significant differences (Fig. 3) between the FRET from DNA (either T9 or T29) to FL and ∆N33-APE1. In the labelling procedure we follow conditions that have been described (Toseland, 2013, ref. 23) to promote specific labelling of the N-terminus, thus we assume that the Cy3 is attached to residue 1 or 34 of APE1, respectively.

  1. On page8 lines 287-289, how to calculate the increase of the factor of 20%?

The average value of acceptor / donor ratio in doubly labelled APE1 for the three conditions (absence of DNA, presence of product or substrate DNA) is 2.363 in the absence of NPM1, as compared to 2.845 in the presence of NPM1 (Fig. 5B), which means an increase of 20.4%.

 9. What is the novel finding of the Effects of NPM1 on APE1/DNA interaction from this manuscript to the References 19, which the authors published in 2020. 

In our article of 2020 (now ref. 20) we described that the NPM1 seems to favour specificity of APE1 for abasic sites (reducing off-target associations) and increase the affinity for substrate DNA, based on biochemical assays. In contrast, in the present study we seek to understand the effect of NPM1 on APE1 N-terminal tail at the molecular /structural level. Although the results are not enough to explain the regulatory mechanism exerted by NPM1, they provide some clue about the conformational rearrangements that might be involved.

The other part also needs to be improved:

  1. The introduction needs to be re-written please update the reference and organize it.

We have re-written the Introduction section, especially the first part. We have included two additional, more recent references for Introduction (4 and 22), and yet another one about AlphaFold and intrinsically disordered protein regions (29).

  1. The first paragraph of the introduction simply introduces the BER pathway, however, it needs to make more clear about the BER pathway, such as, in the short-patch BER pathway, how XRCC1 works with polymerase beta and Ligase III to finish the BER pathway. In addition, what is the long-patch BER pathway? Please update the references for this part.  DNA polymerase β is not DNA polymerase b.

The paragraph has been rewritten to better explain the BER pathway and the role of XRCC1. The long-patch BER subpathway is now mentioned. One very recent review about APE1 and BER has been added to the references. We have checked the spelling of polymerase beta through the text.

  1. In the second paragraph, the endonuclease activity of APE1 is the activity of APE1 incision of the phosphodiester bond with the AP site. To remove the 3’-OH end is APE1 exonuclease activity. So please make this clearly.

This paragraph has been rewritten to explain the exonuclease activity more clearly.

  1. “Thanks to “… is not used in scientific reports.

This expression, which was used twice, has been removed, rewriting the corresponding sentences.

  1. What is  “avidly tetrahydrofurane”?

The sentence has been rewritten for the sake of clarity: “Recombinant APE1 binds with high affinity dumbbell-shaped oligonucleotides containing tetrahydrofurane (THF),...”

  1. In the method part, there is no thermal experiment, the DNA labeling describe. 

The thermal denaturation is followed by circular dichroism (CD), measuring the elipticity at 222 nm as a function of temperature. The details of the experiment have been completed in this section of M&M.

We purchase the oligos already labelled from IDT company. The source (manufacturer) and position of the labels are stated in M&M. We refer to supplementary Fig. S5 to see the labelling positions in the context of the oligos.

Round 2

Reviewer 1 Report

The authors have addressed all of my concerns.

Author Response

Thank you

Reviewer 2 Report

The author improved the manuscript from the revised version, but it still has some questions that need clarification.

Comments:

1.     In figure S1, the author added the marker of the SDS-page, but even with the marker, it does not show the difference between APE1-WT and ∆N33-APE1, especially since it has about a 3 kDa difference in the molecular weight. The author can run the sample on the same SDS-page and compare the size. 

2.     The author only changes the Figure 2 legend, and deletes “Cartoon”, however, the author does not change the figure. Please revise all the figures, use only stick mode to show the DNA.

3.     It is still very confused about the Cys3 labelling. 1st, ref.23(Toseland,2013) does not mention the Cys3 labelling N-ter method. 2nd, in the manuscript, the author mention that the DOL of Cys3-APE1 and Cys3- ΔN33APE1 are 0.9 and 1.7 separately, if the labelling targets the N-ter region, why the truncation shows a high degree of the labelling, then which residues are labelled?

Author Response

  1. In figure S1, the author added the marker of the SDS-page, but even with the marker, it does not show the difference between APE1-WT and ∆N33-APE1, especially since it has about a 3 kDa difference in the molecular weight. The author can run the sample on the same SDS-page and compare the size.

We have analyzed in the same SDS-PAGE all of the APE1 samples under study. The gel is shown in new Fig. S1. The full-length and truncated proteins are run side-by-side, and the migration difference can be clearly seen now.

  1. The author only changes the Figure 2 legend, and deletes “Cartoon”, however, the author does not change the figure. Please revise all the figures, use only stick mode to show the DNA.

We have replaced the figures showing the structural models (Fig. 1, 4 and S6) representing the DNA only with sticks.

  1. It is still very confused about the Cys3 labelling. 1st, ref.23(Toseland,2013) does not mention the Cys3 labelling N-ter method. 2nd, in the manuscript, the author mention that the DOL of Cys3-APE1 and Cys3- ΔN33APE1 are 0.9 and 1.7 separately, if the labelling targets the N-ter region, why the truncation shows a high degree of the labelling, then which residues are labelled?

As mentioned in Toseland et al. (2013) "it is possible to specifically target the N-terminal α-amino group due to a lower pKa of 7 compared to the amino groups of lysine (pKa 10–11)" (page 90 of the journal; 6th page of the article). This α-amino group preferential labelling at pH 7.0 is also stated in Nanda and Lorsch (Methods Enzymol. 2014, 536, p.87) and experimentally documented in Sèlo et al (J Immunol Methods. 1996, 199(2), p.127). These papers do not focus particularly on Cy3 probe, but in general deal with NHS-conjugated fluorophores, such as the NHS-Cy3 we have used. Anyway, we have changed the reference in the manuscript, citing now the more detailed Nanda and Lorsch, 2014.

The preference for amino terminal labelling at neutral pH is not strict and some variability between labelling reactions is not unusual. It is known that lysine residues can exhibit a large range of accessibilities, and that this property varies strongly as a function of protein conformation and the presence of disordered domains and aggregated populations (Lins, L., Thomas, A. & Brasseur, R. Analysis of accessible surface of residues in proteins. Protein Sci 12, 1406–1417, 2003). In this context, it is not entirely surprising to obtain a different DoL for the truncated ΔN33APE1 with partial labelling of some other lysine residue(s) in addition to the amino terminus and suggests an increased accessibility compared to full-length APE1. Variations in DoL ranging from ~10 to 30% for truncated variants compared to full-length APE1 using a similar N-terminus targeting strategy but different N-succinimidyl ester dyes (carboxyfluorescein and carboxytetramethylrhodamine) have been previously reported (Int. J. Mol. Sci. 2020, 21(9), 3122; https://doi.org/10.3390/ijms21093122). In the context of our work, it is important to note that i) by using the acceptor-sensitized method to calculate the FRET efficiencies, which uses the ratio of acceptor intensities obtained through FRET and via direct excitation, the impact of different DoL between full-length and truncated versions is removed, and ii) truncated APE1 displayed a similar trend as the full-length APE1 with the N-terminus being closer to the T9 than T29, thus suggesting that the FRET partners and vector is identical in both species.

Furthermore, we do not think that the different DoL between full-length and ΔN33APE1 is due to any intrinsic difference between the proteins but to the reasons exposed above and/or fortuitous variability among reactions (in spite of our efforts to reproduce the conditions). Thus, although the specific labelling at only one residue is always pursued, sometimes one gets over-labelling as in the case of DoL 1.7. To further clarify this issue we have added a sentence in this paragraph of section 2.1, warning about this uncertainty.